# Generating Fine-Scale Aerosol Data through Downscaling with an Artificial Neural Network Enhanced with Transfer Learning

Menglin Wang [1,†], Meredith Franklin [1,2,*,†] and Lianfa Li [1,3]

1. Division of Biostatistics, University of Southern California, Los Angeles, CA 90032, USA; menglinw@usc.edu (M.W.); lianfali@usc.edu (L.L.)
2. Department of Statistical Sciences and School of the Environment, University of Toronto, Toronto, ON M5G 1Z5, Canada
3. State Key Laboratory of Resources and Environmental Information System, Institute of Geographical Sciences and Natural Resources, Chinese Academy of Sciences, 11A, Datun Road, Chaoyang District, Beijing 100101, China
* Correspondence: meredith.franklin@utoronto.ca
† These authors contributed equally to this work.

**Abstract:** Spatially and temporally resolved aerosol data are essential for conducting air quality studies and assessing the health effects associated with exposure to air pollution. As these data are often expensive to acquire and time consuming to estimate, computationally efficient methods are desirable. When coarse-scale data or imagery are available, fine-scale data can be generated through downscaling methods. We developed an Artificial Neural Network Sequential Downscaling Method (ASDM) with Transfer Learning Enhancement (ASDMTE) to translate time-series data from coarse- to fine-scale while maintaining between-scale empirical associations as well as inherent within-scale correlations. Using assimilated aerosol optical depth (AOD) from the GEOS-5 Nature Run (G5NR) (2 years, daily, 7 km resolution) and Modern-Era Retrospective analysis for Research and Applications, Version 2 (MERRA-2) (20 years, daily, 50 km resolution), coupled with elevation (1 km resolution), we demonstrate the downscaling capability of ASDM and ASDMTE and compare their performances against a deep learning downscaling method, Super Resolution Deep Residual Network (SRDRN), and a traditional statistical downscaling framework called dissever ASDM/ASDMTE utilizes empirical between-scale associations, and accounts for within-scale temporal associations in the fine-scale data. In addition, within-scale temporal associations in the coarse-scale data are integrated into the ASDMTE model through the use of transfer learning to enhance downscaling performance. These features enable ASDM/ASDMTE to be trained on short periods of data yet achieve a good downscaling performance on a longer time-series. Among all the test sets, ASDM and ASDMTE had mean maximum image-wise $R^2$ of 0.735 and 0.758, respectively, while SRDRN, dissever GAM and dissever LM had mean maximum image-wise $R^2$ of 0.313, 0.106 and 0.095, respectively.

**Keywords:** downscaling; artificial neural network; transfer learning; deep learning; G5NR; MERRA-2

## 1. Introduction

Fine-scale aerosol data provide essential support for air quality studies [1] and downstream health-related applications. Over the past several years, satellite-based aerosol optical depth (AOD) has been used for this purpose, primarily to estimate $PM_{2.5}$ surfaces at fine spatial scales [2–4]. Satellite AOD-derived $PM_{2.5}$ estimates have been used to examine health outcomes including respiratory [5–7] and cardiovascular [8] diseases. Generating fine-scale $PM_{2.5}$ from satellite AOD has several limitations including missing data due to cloud cover and bright surfaces [9], and it requires complex statistical or machine learning techniques that incorporate multiple external data sources [10].

Our study region encompasses several countries across Southwest Asia (Afghanistan, Iraq, Kuwait, Saudi Arabia, United Arab Emirates, and Qatar (Figure 1)), which is known

for its extreme dry and hot hyper-arid climate. This unique environment, in addition to increased economic development and urbanization, makes both naturally and anthropogenically occurring air pollution a concern [11]. This region is also the basis of a larger research initiative assessing the impact of air quality on the health of military personnel that were deployed in the region during post 9/11 wars [12,13]. As there is very little ground-level air quality monitoring in the region, having fine-scale aerosol data is an asset to support air pollution related research.

Recent advances in data assimilation products provide a source of AOD data, with the NASA Modern-Era Retrospective Analysis for Research and Applications, version 2 (MERRA-2) drawing intensive research interest since it provides complete surfaces of AOD and related aerosol products globally from 1980 onward. Given its long time range of available data, Sun et al. (2019) [14] analyzed the spatial distribution and temporal variation of MERRA-2 AOD over China from 1980 to 2017. Ukhov et al. (2020) [15] used MERRA-2 AOD to assess natural and anthropogenic air pollution over the Middle East. However, the spatial resolution of MERRA-2 data is quite coarse (~50 km) which limits its application for local-scale research. In contrast, the Goddard Earth Observing System Model, Version 5 (GEOS-5) Nature Run (G5NR) [16] can provide AOD data in finer resolution (~7 km). G5NR is a global non-hydrostatic mesoscale simulation performed by the GEOS-5 Atmospheric General Circulation Model (GEOS-5 AGCM) and driven by prescribed sea-surface temperature, sea ice, surface emissions and uptake of aerosols and trace gases [16]. However, G5NR is only available for two-years (2005–2007) due to its high computational cost which restricted its research potential.

Statistical downscaling from coarse- to fine-scale is a computationally efficient solution to generate fine-scale aerosol data, which can take advantage of both the long temporal range of MERRA-2 AOD and the fine spatial resolution of G5NR AOD. Statistical downscaling was developed primarily to generate finer spatial scale climate information from General Circulation Models (GCMs) [17], and these techniques have also been applied to remote sensing data [3,18–21]. The basic approach to statistical downscaling is that the fine (smaller) scale variable is conditioned by a coarse (larger) scale variable and local features, like topography and land-sea distribution [22]. In this perspective, the fine-scale variable can be predicted with an empirical association that relates the coarse-scale variables (predictors) and fine-scale variables (predictands). For instance, dissever is a general framework for downscaling earth resource information [18]. It uses an iterative algorithm to fit regression models between coarse- and fine-scale variables in order to optimize downscaling by ensuring the value of each coarse grid is equal to the mean of fine-scale values that are spatially covered by the corresponding coarse grid.

Deep learning [23] has surpassed traditional statistical approaches with considerable performance improvements, and has thus been used in a variety of remote sensing data applications [24]. The convolutional neural network (CNN) is a popular method for downscaling due to its ability to learn spatial features from large gridded data [25]. Recently, a CNN-based model called the Super Resolution Deep Residual Network (SRDRN), which utilized convolutional layers and residual networks, was developed to downscale daily precipitation and temperature [26]. Autoencoder-like models with residual connections and parameter sharing have also been used to downscale by incorporating an iterative training strategy to force spatial value consistency [27]. Networks with transfer learning have been used in a spatial context to generalize the empirical associations within one region to apply downscaling in a different region, showing notable improvement compared to classical statistical downscaling methods [26,28].

We propose an artificial neural network (ANN) [29] sequential downscaling method (ASDM) with transfer learning enhancement (ASDMTE). ASDM/ASDMTE utilizes empirical between-scale associations, and accounts for inherent within-scale temporal associations among fine-scale data. In addition, within-scale temporal associations in the coarse-scale data being downscaled are integrated into the ASDMTE model through the use of transfer learning to enhance downscaling performance.

Under the ASDM framework, the fine-scale variable can be modeled as a non-linear function of coarse-scale variable, with a sequence of temporally lagging fine-scale variables at the same location adjusting for geographic information (e.g., elevation), time (day of the year) and location (latitude, longitude). To enhance the performance of ASDM, transfer learning can be incorporated where another similar sequential ANN model is trained on the long time series of coarse-scale data to learn its inherent temporal associations; this model is then transferred into ASDM to enhance its downscaling performance.

We developed ASDM/ASDMTE models to downscale AOD data obtained from the Modern-Era Retrospective analysis for Research and Applications, Version 2 (MERRA-2), a satellite-based reanalysis product produced by NASA's Global Modeling and Assimilation Office (GMAO). MERRA-2 data are available for a long period (1980–present) at relatively coarse scale (~50 km). The target for downscaling was fine-scale (~7 km) AOD from the Goddard Earth Observing System Model, Version 5 (GEOS-5) Nature Run (G5NR), another satellite-based product [16]. At this resolution, G5NR is an informative data source for understanding local-scale air quality and as an exposure metric for health effects studies, but it is limited in temporal range (2005–2007), which restricts its broad use for long-term studies. As the fine-scale G5NR data has limited temporal range (2 years of daily data), it was difficult to build stable empirical associations needed for traditional statistical downscaling that link large-scale variables with local-scale variables. Furthermore, little external or covariate information were available at fine scales that could help with traditional downscaling. These limitations made it impractical to establish between-scale empirical associations without other prior knowledge, particularly since the single coarse-scale variable did not have enough spatial variability to predict the fine-scale variable. Lastly, even though G5NR and MERRA-2 provide the same variables over the same region and period of time, they are independent datasets that do not match on a point-to-point basis due to algorithmic differences [16]. Specifically, the mean of the G5NR 7 km grid values is not exactly equal to its coincident MERRA-2 50 km coarse grid value.

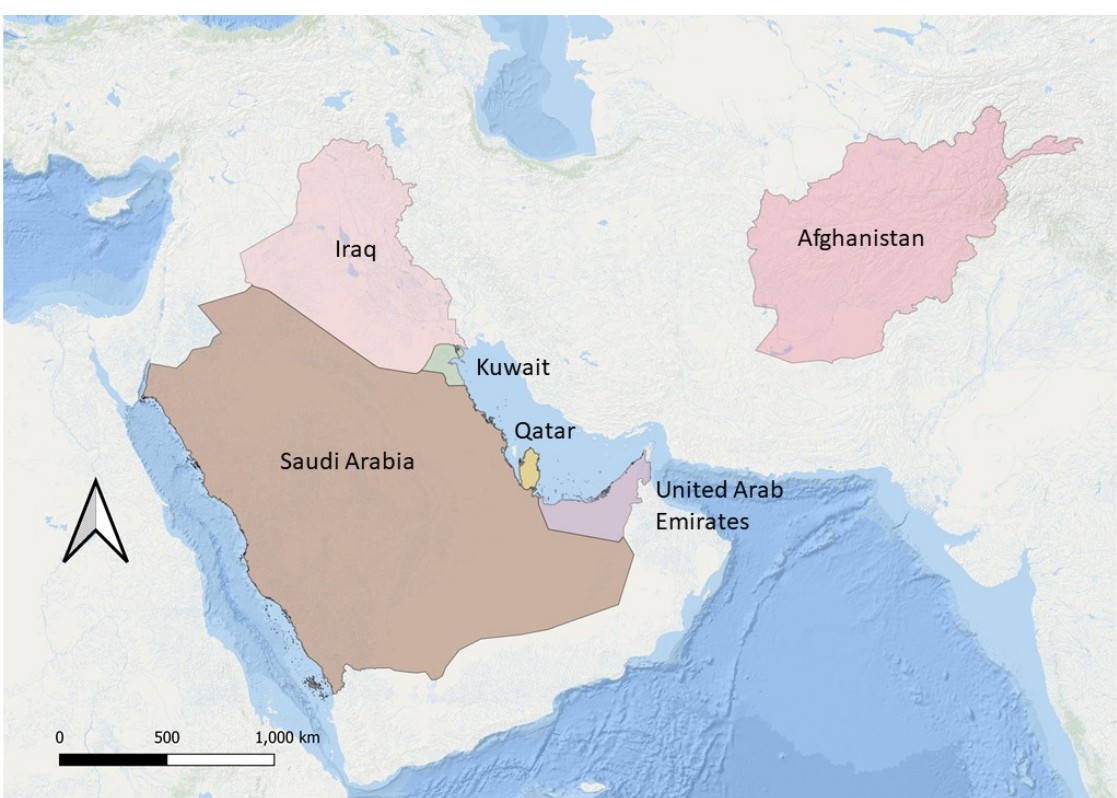

**Figure 1.** Map of the study domain.

We applied our ASDM and ASDMTE downscaling approaches to G5NR and MERRA-2 data for several countries in Southwest Asia (Figure 1). ASDM/ASDMTE performances were compared with a deep learning downscaling method, Super Resolution Deep Residual Network (SRDRN) and a traditional statistical downscaling methods in the dissever framework including generalized additive models (GAM), and linear regression model (LM) over the same study domain and period.

## 2. Materials and Methods

### 2.1. Data

#### 2.1.1. MERRA-2

The Modern-Era Retrospective Analysis for Research and Applications, version 2 (MERRA-2) is a multi-decadal atmospheric reanalysis product produced by NASA's Global Modeling and Assimilation Office (GMAO) [30]. Using the Goddard Earth Observing System, version 5 (GEOS-5) [30], of which the key components are an atmospheric model [31,32] and Gridpoint Statistical Interpolation (GSI) analysis scheme [33,34], MERRA-2 assimilates AOD from various ground- and space-based remote sensing platforms [35] and uses an aerosol module to simulate 15 externally aerosol mass mixing ratio tracers [36]. We used Total Aerosol Extinction AOD 550 nm (AOD) [37]. While the MERRA-2 data are available from 1980 forward, our study period was 16 May 2000–15 May 2018. MERRA-2 AOD data has 0.625° longitudinal resolution, 0.5° latitudinal resolution (∼50 km) and daily temporal resolution.

#### 2.1.2. G5NR

GEOS-5 Nature Run (G5NR) is a two-year (16 May 2005–15 May 2007) non-hydrostatic 7 km global mesoscale simulation also produced by the GEOS-5 atmospheric general circulation model [38]. Its development was motivated by the observing system simulation experiment (OSSE) community for a high-resolution sequel to the existing Nature Run, European Centre for Medium-Range Weather Forecasts (ECMWF). Like MERRA-2, G5NR includes 15 aerosol tracers [16]. It simulates its own weather system around the Earth which is constrained only by surface boundary conditions for sea-surface temperatures, the burning emissions of sea-ice, daily volcanic and biomass and high-resolution inventories of anthropogenic sources [38]. In this study we focused on all two years of the available G5NR Total Aerosol Extinction AOD 550 nm, which had 0.0625° grid resolution (∼7 km) and daily temporal resolution.

#### 2.1.3. GMTED2010 Elevation

The Global Multi-resolution Terrain Elevation Data 2010 (GMTED2010) is a global elevation model developed by the U.S. Geological Survey and the National Geospatial-Intelligence Agency [39]. The data are available at three separate resolutions (horizontal post spacing) of 30 arc-seconds (∼1 km), 15 arc-seconds (∼500 m), and 7.5 arc-seconds (∼250 m) [40]. We used the 30 arc-seconds resolution data and spatially averaged to match the ∼7 km G5NR grid.

### 2.2. Downscaling Model

We propose an Artificial Neural Network Sequential Downscaling Method (ASDM) with Transfer Learning Enhancement (ASDMTE) to generate fine-scale (FS) data from coarse-scale (CS) data. The method can be formulated as follows:

Let $y_{i,j,t}$ denote the FS AOD referenced at $i, j, t$, where $i \in \{1, 2, \cdots, h\}$, $j \in \{1, 2, \cdots, w\}$, $t \in \{1, 2, \ldots, d\}$; $h$ and $w$ index latitude and longitude over the study domain and $d$ is the time index. Similarly, we define the CS AOD referenced at $x_{i',j',t'}$, where $i' \in \{1, 2, \cdots, h'\}$, $j' \in \{1, 2, \cdots, w'\}$, $t' \in \{1, 2, \cdots, d'\}$; $h'$, $w'$ and $d'$ are latitude, longitude and time indices, respectively. Although the CS data have a longer overall period of temporal coverage, the FS and CS data have the same time step (day).

The estimated downscaling model $\hat{f}$ can then be denoted as:

$$y_{i,j,t} = \hat{f}(\boldsymbol{y_{(i,j,t-1),n}}, x_{i',j',t}, Ele_{i,j}, Lat_i, Lon_j, Day_t)$$

$$\boldsymbol{y_{(i,j,t-1),n}} = y_{i,j,t-1}, \cdots, y_{i,j,t-n},$$

(1)

where $Ele_{i,j}, Lat_i, Lon_j, Day_t$ are elevation, latitude, longitude and day of the year at $i, j, t$, respectively; $x_{i',j',t}$ represents CS AOD that spatially covers $y_{i,j,t}$ (at the same time t); $\boldsymbol{y_{(i,j,t-1),n}}$ is a list of $n$ temporal lagging variables at location $i, j$.

Through $\hat{f}$, we not only learned empirical associations between the CS and FS variables, $x_{i',j',t}$ and $y_{i,j,t}$, but also short-term temporal associations within the FS data by including $n = 25$ time lags of the fine-scale variables, $\boldsymbol{y_{(i,j,t-1),n}}$. In the model we also adjusted for location (latitude, $Lat_i$ and longitude, $Lon_j$), long-term time (day of the year, $Day_t$), and geographic information (elevation, $Ele_{i,j}$) making $\hat{f}$ a function of space and time. This also enabled the use of data at different locations and times to train our model, which provided more information for training and partially alleviated the issue of having limited overlapping (in time) data. The larger the spatial area and temporal range, the more data we had for training; however, at the same time, the model $\hat{f}$ became more complex. This increasing complexity in the target model is equivalent to adding difficulty in the learning process, thus we made the decision to trade off between data availability and model complexity.

To enhance the performance of $\hat{f}$, we incorporated transfer learning [41] into ASDM. Machine learning methods traditionally solve isolated tasks from scratch, which make them data hungry. Transfer learning attempts to solve this problem by developing methods to transfer knowledge learned in other sources and use it to improve the learning performance in a related target task [42]. The formal definition of transfer learning can be expressed as [41]:

**Definition 1** (Transfer Learning). *Given a source domain $\mathcal{D}_S$ and learning task $\mathcal{T}_S$, a target domain $\mathcal{D}_T$ and learning task $\mathcal{T}_T$, transfer learning aims to help improve the learning of the target predictive function $h_T(\cdot)$ in $\mathcal{D}_T$ using the knowledge in $\mathcal{D}_S$ and $\mathcal{T}_S$, where $\mathcal{D}_S \neq \mathcal{D}_T$, or $\mathcal{T}_S \neq \mathcal{T}_T$.*

Transfer learning allows us to learn certain patterns within one dataset that can be applied to another. Since coarse-scale data are usually cheaper to obtain and more available, we can use inherent knowledge learned within them to improve the predictive performance of $\hat{f}$. Thus, to make use of the spatiotemporal associations within the CS data, a transfer model was trained on CS data to learn the inherent mapping function $\hat{g}$ and, consequently, the model $\hat{g}$ was transferred into the ASDM/ASDMTE. The transfer integration of the ASDMTE network structure is shown in Figure 2. The learned inherent function $\hat{g}$ can be denoted as:

$$x_{i',j',t'} = \hat{g}(\boldsymbol{x_{(i',j',t'-1),n}})$$

$$\boldsymbol{x_{(i',j',t'-1),n}} = x_{i',j',t'-1}, \cdots, x_{i',j',t'-n}.$$

(2)

### 2.2.1. ASDM/ASDMTE Network Structure

Given its ability to fit non-linear functions, we used an artificial neural network to model $\hat{f}$; the overall network structure of ASDM/ASDMTE is shown in Figure 2.

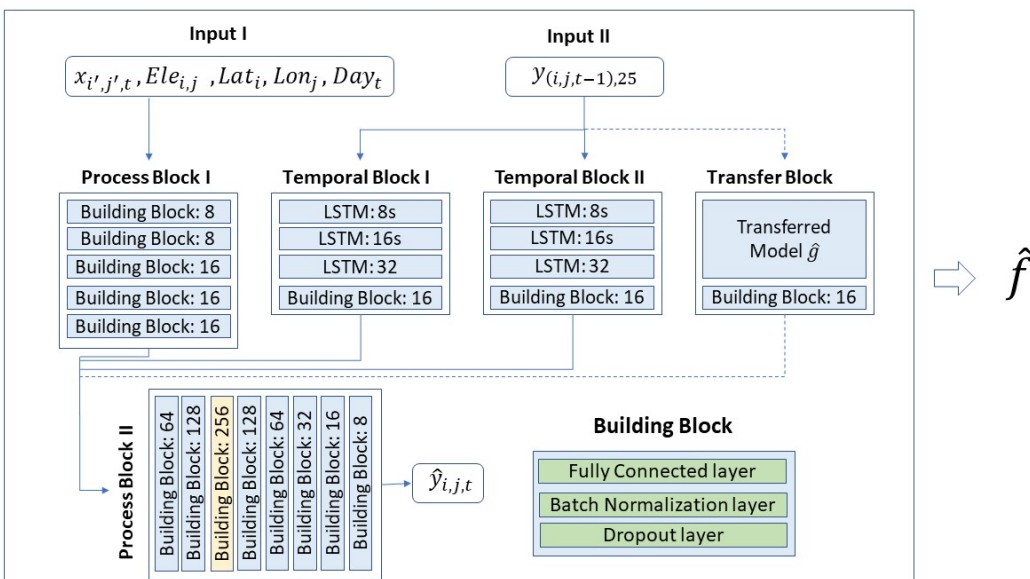

**Figure 2.** Overall Neural Network structure of ASDM/ASDMTE. The notation LSTM:8s represents a LSTM layer with 8 nodes and return sequence. The notation of Building Block:8 represents a building block with 8 nodes. The light yellow block represents using dropout layer with dropout rate of 0.5. The transfer Block is only used in ASDMTE thus it is connected with dash lines.

For model fitting, longitude, latitude, day of the year and elevation were normalized to a range $[0, 1]$. The CS and FS AOD variables, $X, Y$, have natural range $[0, 6]$ which is approximately the same scale as $[0, 1]$ and thus they were kept on their original scale. 'Input I' used all available features except lagging variables, $X_{i', j', t}, Ele_{i, j}, Lat_i, Lon_j, Day_t$, and was processed by 'Process Block I'. 'Input II' was composed of the 25 FS lags $y_{(i, j, t-1), 25}$ and went through 'Temporal Block I' and 'Temporal Block II' in ASDM. If using transfer learning enhancement (ASDMTE), 'Input II' was also processed by the 'Transfer Block'. All output from 'Process Block I', 'Temporal Block I', 'Temporal Block II' and/or 'Transfer Block' were combined and then processed by 'Process Block II'.

Long Short Term Memory (LSTM) [43] was used to model the within scale temporal associations. The building block of ASDM/ASDMTE was composed of a fully connected (FC) layer, a batch normalization layer, and an optional dropout layer. Leaky ReLU [44] was used as a non-linear activation function of the FC layer to prevent dead neurons and can be expressed as:

$$\text{LeakeyReLU}(x) = \begin{cases} x & \text{if } x > 0 \\ \alpha x & \text{otherwise,} \end{cases}$$

where we chose $\alpha = 0.1$. The batch normalization layer was used to stabilize the learning process and reduce the training time [45,46]. Dropout layers with rate 0.5 were used as regularization to prevent overfitting [47,48], but the dropout layer was applied only in selected building blocks, marked in yellow in Figure 2. The loss function of this model was Mean Square Error (MSE), which can be expressed as:

$$MSE = \frac{1}{n} \sum_{i=1}^{n} (Y_{i, j, t} - \hat{Y}_{i, j, t}). \tag{3}$$

### 2.2.2. Transferred Model

The transferred model was trained on CS data (MERRA-2), resulting in the learned function $\hat{g}$ (Equation (2)). Its network structure is shown in Figure 3.

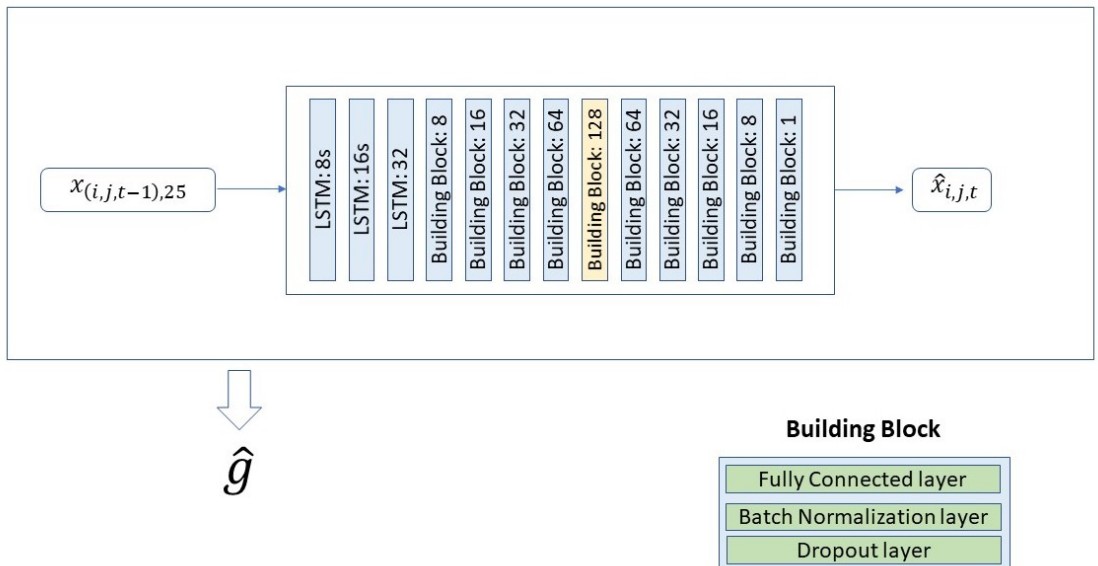

**Figure 3.** Neural Network structure of the transferred model. The notation LSTM: 8s represents a LSTM layer with 8 nodes and return sequence. The notation of Building Block:8 represents a building block with 8 nodes. The light yellow block represents the dropout layer with dropout rate 0.5.

The transferred model captured the within-scale association in CS data and carries this spatiotemporal knowledge to the ASDM to enhance its performance. The Neural Network used to learn $\hat{g}$ was composed of the same building block and similar structure as ASDM/ASDMTE. We used mean squared error (MSE) as the loss function, and to prevent overfitting, dropout layer and early stopping training were applied. We randomly chose 10% of available days as the validation set for early stopping. The 'Transferred Model' is integrated as part of ASDMTE network directly by setting it to untrainable (i.e., it was not updated during training).

### 2.2.3. Training Strategy

There is always a trade-off between model complexity and data size. The larger spatial and temporal coverage of the data used for training, the more complex the target function $f$ becomes. As this makes it more difficult to learn, we simplified the learning task by spatially and temporally splitting the data while maintaining a reasonable data size, and fitting separate models on each of the subsets. Spatially, the data were grouped into four regions: 1. Afghanistan; 2. United Arab Emirates and Qatar; 3. Saudi Arabia; and 4. Iraq and Kuwait. Temporally, the data were divided approximately equally into four seasons that have 91, 91, 91 and 92 days, respectively. In order to produce temporally continuous downscaled predictions, a 45-day overlap was added to each season as shown in Figure 4.

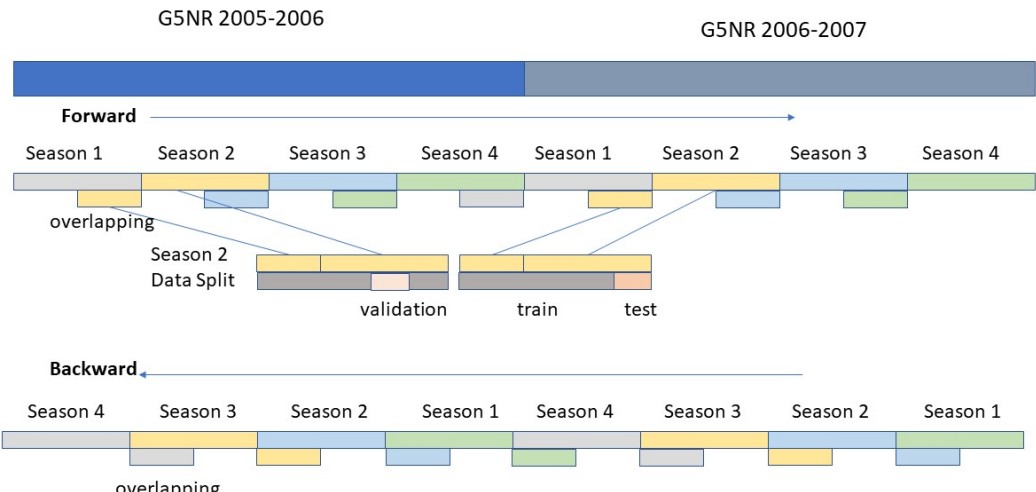

**Figure 4.** Temporal simplification and splitting for forward and backward prediction.

The model in Equation (1) illustrates the prediction for the forward temporal direction; that is, to predict the future with historical observations. We also trained a backward prediction model with a slight variation of the same model format, but using future observations to predict historical data (Figure 4). Training this way allowed downscaling in both directions, forward and backward in time, which was needed for our application where we aimed to downscale before and after the 2-year training period. Consequently, 32 models (4 regions × 4 seasons × 2 directions) were fitted on all combinations of region, season and direction. Within each subset of data, the data were composed of the same seasons from two years (2005 and 2006), as shown in Figure 4. The two years of data were evenly divided into 10 parts and the last 10% of the data were used as test set. The validation set was the fourth 10% of data. The remaining 80 % was used as the training set.

2.2.4. Evaluation

The downscaling results in the same direction and time were combined spatially as whole images for evaluation purposes. The main evaluation metrics were image-wise $R^2$ [18] and Root Mean Square Error (RMSE), which are defined as follows:

$$R_t^2 = 1 - \frac{\sum_{i=1}^{h} \sum_{j=1}^{w} (y_{i,j,t} - \hat{y}_{i,j,t})^2}{\sum_{i=1}^{h} \sum_{j=1}^{w} (y_{i,j,t} - \bar{y}_t)^2}$$

$$RMSE_t = \sqrt{\frac{1}{hw} \sum_{i=1}^{h} \sum_{j=1}^{w} (y_{i,j,t} - \hat{y}_{i,j,t})^2}, \tag{4}$$

where $\hat{y}_{i,j,t}$ is the downscaled AOD value at $i,j,t$ and $y_{i,j,t}$ is the corresponding true value. The downscaled results of ASDM, ASDM with transfer enhancement (ASDMTE), SRDRN, dissever framework with GAM and LM as regressors were compared on the same test sets with the above metrics. The structure of SRDRN can be found in Wang et al. (2021) [26].

**3. Results**

Same-day images of the 7 km G5NR and 50 km MERRA-2 images are shown in Figure 5. We note similarities in their spatial trends with higher values in arid regions of southeast Saudi Arabia and United Arab Emirates (UAE), but greater definition in the fine scale G5NR image that is particularly clear over Afghanistan. The bottom left and bottom right plots of Figure 5 show mean image-wise $R^2$ and RMSE (respectively) of G5NR and MERRA-2 AOD data with different lagging. Both G5NR and MERRA-2 show similar temporal associations: the further two data images are, the less they are associated,

indicated by lower image-wise $R^2$ and higher RMSE. These similar inherent temporal associations of G5NR and MERRA-2 provided a good foundation for ASDM to assume that local-scale AOD can be predicted not only by between-scale associations, but also by inherent within-scale associations. In addition, due to generative algorithm differences between G5NR and MERRA-2 AOD data, G5NR AOD has a universally higher mean value and standard deviation (0.316 (0.258)) compared to MERRA-2 AOD (0.294 (0.197)), which is the reason G5NR had higher lagging *RMSE* and $R^2$ (Figure 5).

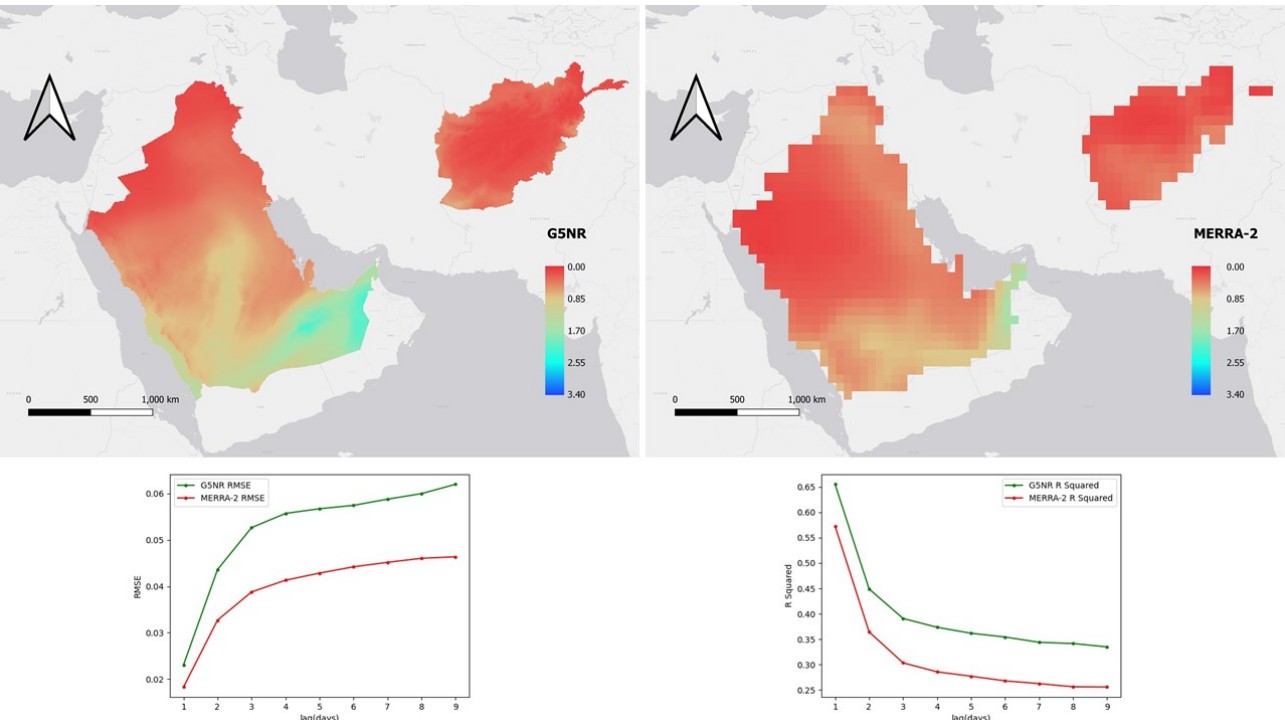

**Figure 5.** Sample images from G5NR (**top-left**) and MERRA-2 (**top-right**) on 29 July 2006; temporal trend of image-wise RMSE (**bottom-left**) and $R^2$ (**bottom-right**) with different lags (days).

Model performance results comparing ASDM and ASDMTE against SRDRN, GAM and LM are shown in Appendix Figures A1–A4. Both ASDM and ASDMTE outperformed other methods, as indicated by higher image-wise $R^2$ and lower RMSE across all seasons and directions. Among all of the test sets, ASDMTE had average maximum image-wise $R^2$=0.758 and average mean image-wise $R^2$=0.443. The ASDM performed similarly with average maximum image-wise $R^2$=0.735 and average mean image-wise $R^2$=0.431. The SR-DRN, GAM and LM methods had average maximum image-wise $R^2$=0.313, 0.106 and 0.095 respectively. Notably, the downscaled AOD map generated by ASDMTE and ASDM on 29 July 2006 (Figure 6a,b) preserved very similar spatial characteristics as the true G5NR data in Figure 6d, while SRDRN and dissever-based downscaling results (see Figure 6c,e,f) exhibit clearly different patterns.

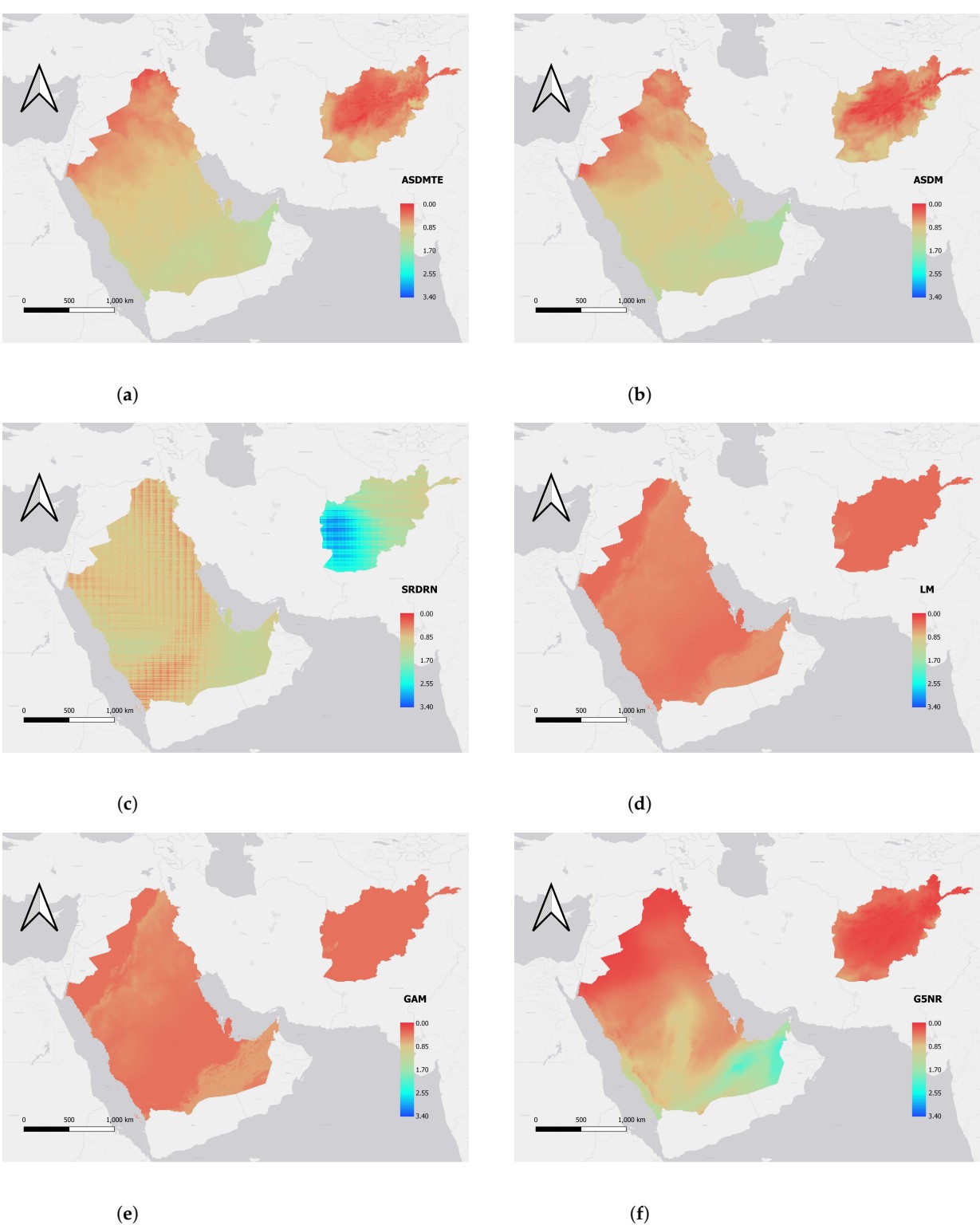

**Figure 6.** Downscaled (by method) and G5NR data over the study region on 29 July 2006: (**a**) ASDMTE, (**b**) ASDM, (**c**) SRDRN, (**d**) dissever GAM, (**e**) dissever LM, (**f**) G5NR.

## 4. Discussion

In this study we developed an Artificial Neural Network Sequential Downscaling Method (ASDM) with Transfer Learning Enhancement (ASDMTE) that enabled coarse-scale AOD data (~50 km) to be downscaled to a finer-scale (~7 km) where training occurred only on a limited sample of temporally overlapping images. The ASDM/ASDMTE approach took point-wise inputs of lagged fine-scale AOD data, coarse-scale AOD data, latitude,

longitude, time and elevation to predict the fine-scale AOD generated from G5NR. We found that this neural network approach was able to learn complex relationships and produce reliable predictions. Based on the comparison of image-wise $R^2$ and *RMSE* shown in Appendix Figures A1–A4 and Table 1, ASDM/ASDMTE showed superior downscaling performance that outperformed the CNN-based neural network—SRDRN—and statistical downscaling approaches in dissever (GAM, LM).

**Table 1.** Image-wise $R^2$ and RMSE from downscaling (by method); $R^2$ is presented as Max (Mean), and RMSE is presented as Mean (SD).

| Method | | Mean | Forward | | | | Backward | | | |
|---|---|---|---|---|---|---|---|---|---|---|
| | | | Season 1 | Season 2 | Season 3 | Season 4 | Season 1 | Season 2 | Season 3 | Season 4 |
| ASDMTE | $R^2$ | 0.758 (0.443) | 0.857 (0.593) | 0.831 (0.381) | 0.728 (0.396) | 0.628 (0.174) | 0.595 (0.360) | 0.851 (0.496) | 0.802 (0.653) | 0.770 (0.488) |
| | RMSE | 0.067 (0.021) | 0.051 (0.010) | 0.061 (0.013) | 0.074 (0.017) | 0.058 (0.014) | 0.088 (0.018) | 0.069 (0.014) | 0.043 (0.005) | 0.094 (0.075) |
| ASDM | $R^2$ | 0.735 (0.431) | 0.890 (0.656) | 0.810 (0.371) | 0.642 (0.394) | 0.588 (0.185) | 0.576 (0.313) | 0.851 (0.415) | 0.790 (0.616) | 0.732 (0.494) |
| | RMSE | 0.068 (0.020) | 0.045 (0.008) | 0.062 (0.013) | 0.077 (0.012) | 0.057 (0.012) | 0.089 (0.016) | 0.064 (0.010) | 0.045 (0.007) | 0.106 (0.078) |
| SRDRN | $R^2$ | 0.313 (0.088) | 0.425 (0.177) | 0.198 (0.067) | 0.268 (0.063) | 0.422 (0.123) | 0.239 (0.075) | 0.211 (0.040) | 0.412 (0.094) | 0.332 (0.067) |
| | RMSE | 0.088 (0.083) | 0.177 (0.131) | 0.067 (0.060) | 0.063 (0.060) | 0.123 (0.108) | 0.075 (0.067) | 0.040 (0.046) | 0.094 (0.098) | 0.067 (0.098) |
| dissever GAM | $R^2$ | 0.106 (0.046) | 0.199 (0.155) | 0.139 (0.055) | 0.056 (0.015) | 0.040 (0.013) | 0.079 (0.018) | 0.070 (0.009) | 0.143 (0.068) | 0.124 (0.038) |
| | RMSE | 0.213 (0.039) | 0.359 (0.058) | 0.130 (0.012) | 0.161 (0.030) | 0.280 (0.055) | 0.172 (0.052) | 0.131 (0.014) | 0.293 (0.045) | 0.181 (0.044) |
| dissever LM | $R^2$ | 0.095 (0.040) | 0.173 (0.133) | 0.108 (0.047) | 0.067 (0.015) | 0.031 (0.013) | 0.087 (0.017) | 0.062 (0.008) | 0.121 (0.048) | 0.113 (0.037) |
| | RMSE | 0.214 (0.039) | 0.362 (0.059) | 0.130 (0.012) | 0.161 (0.031) | 0.279 (0.055) | 0.170 (0.051) | 0.131 (0.013) | 0.295 (0.045) | 0.181 (0.044) |

Statistical downscaling has a long history, rooting from the demand to generate local-scale climate information from GCMs with less computational cost. Traditional statistical approaches focus on establishing empirical associations between coarse-scale and fine-scale variables [22,49]. For instance, Loew et al. (2008) modeled the associations between soil moisture at 40 km resolution and its corresponding fine-scale (1 km) observations using linear regression [50]. Leveraging temporal replicates, they fit separate linear regression models independently to each fine scale grid, ignoring spatial and temporal associations in either the fine- or coarse-scale data. Recently, deep learning approaches have been used that address spatial features, such as Wang et al. (2021) [26], who developed a CNN-based method, Super Resolution Deep Residual Network (SRDRN), to downscale precipitation and temperature from coarse resolutions (25, 50 and 100 km) to fine resolution (4 km) by learning the between-scale image-to-image mapping function. However, they ignore the temporal associations between images.

Current downscaling methods focus only on modeling between-scale relationships and ignore any inherent temporal associations in the data. As observed in Figure 5, there are inherent within-scale temporal associations in the fine- and coarse-scale data, where at the same location temporally near observations tend to be correlated to each other. These associations provided essential support for downscaling and resulted in better fine-scale

predictions. Essentially, the target fine-scale variable can be estimated by the coarse-scale variable as well as its own temporal lagging, adjusting for geographic features, location and time.

By defining the downscaling problem as above, the ASDM/ASDMTE approach was able to take advantage of both the within-scale temporal associations in the fine-scale data, and between-scale spatial associations, which allow it to have more information with which the neural network can learn better than just using the between-scale spatial relationships. This richness in predictive information is especially important in a situation where data are limited, since it can enable the model to be trained on a short period of overlapping data without requiring point-to-point matching of the fine- and coarse-scale images.

This setting also enabled the use of transfer learning (through ASDMTE) by leveraging the within-scale temporal associations in the coarse-scale data, which had a much longer time series. Typically in downscaling only the temporally overlapping coarse- and fine-scale data can be used for modeling. However, in our case we wanted to downscale a longer time series, and we were able to use transfer learning to learn from all (2000–2018) coarse-scale MERRA-2 data by training $\hat{g}$ and transferring it to enhance the downscaling model.

ASDM/ASDMTE suffers from the same assumption of stationarity as other downscaling methods, that is it assumes the statistical association between coarse- and fine-scale data does not change outside of the model training time [51,52]. In addition, we may need to further assume stationary of within-scale temporal associations (i.e., temporal lags) used in the model.

Another concern of ASDM/ASDMTE is its test robustness. To stabilize $\hat{f}$ at test time, we trained different ASDM/ASDMTE models for each season of a year and separately for different regions/countries, as shown in Section 2.2.3. The shorter period of time and smaller target domain simplified the learning task of each model and at the same time, simplified the domain to which the model needed to generalize, so we obtained more robust results when testing.

In addition, ASDM/ASDMTE was designed to solve a supervised downscaling problem, that is, to downscale coarse-scale data and validate against fine-scale data. It requires the presence of some fine-scale data and ASDM/ASDMTE can computationally efficiently extend its temporal range by utilizing the within-scale temporal association to downscale. In the absence of fine-scale data, ASDM/ASDMTE cannot be applied.

A further research direction would be to stabilize the sequential downscaling performance in the presence of shorter temporal range of fine-scale data to account for predicting over a long time series. As shown in Appendix Figures A1–A4, ASDM/ASDMTE can have good downscaling performances and their performances can even recover from previous bad downscaled results, but the performance still shows a temporally decreasing trend. Our future research will focus on improved learning of stable temporal associations to improve sequential downscaling performance for long time series prediction.

**Author Contributions:** Conceptualization, M.F. and M.W.; methodology, M.W.; validation, M.W.; formal analysis, M.W.; resources, M.F.; data curation, M.F.; writing—original draft preparation, M.W., M.F.; writing—review and editing, M.F., L.L.; visualization, M.W.; supervision, M.F.; project administration, M.F.; funding acquisition, M.F. All authors have read and agreed to the published version of the manuscript.

**Funding:** This research was funded by National Aeronautics and Space Administration grant number 80NSSC19K0225.

**Institutional Review Board Statement:** Not applicable.

**Informed Consent Statement:** Not applicable.

**Data Availability Statement:** Analysis data are available upon request.

**Conflicts of Interest:** The authors declare no conflict of interest.

## Abbreviations

The following abbreviations are used in this manuscript:

| | |
|---|---|
| ANN | Artificial Neural Network |
| AOD | Aerosol Optical Depth |
| ASDM | Artificial Neural Network Sequentially Downscaling Method |
| ASDMTE | ASDM with Transfer Learning Enhancement |
| CNN | Convolutional Neural Netwrok |
| CS | Coarse-Scale |
| ECMWF | European Centre for Medium-Range Weather Forecasts |
| FC | Fully Connected |
| FS | Fine-Scale |
| G5NR | GEOS-5 Nature Run |
| GAM | Generalized Additive Model |
| GCM | General Circulation Model |
| GEOS-5 | Goddard Earth Observing System Model, Version 5 |
| GEOS-5 AGCM | GEOS-5 Atmospheric General Circulation Model |
| GMAO | Global Modeling and Assimilation Of-44fice |
| GMTED2010 | The Global Multi-resolution Terrain Elevation Data 2010 |
| LM | Linear Regression Model |
| MERRA-2 | Modern-Era Retrospective analysis for8Research and Applications, Version 2 |
| MSE | Mean Square Error |
| NGA | Geospatial-Intelligence Agency |
| OSSEs | Observing System Simulation Experiments |
| ReLU | Rectified Linear Unit |
| RMSE | Root Mean Square Error |
| SD | Standard Deviation |
| SRDRN | Super Resolution Deep Residual Network |
| USGS | U.S. Geological Survey |
| UAE | United Arab Emirates |

## Appendix A. Supplemental Results: Downscaling Performance

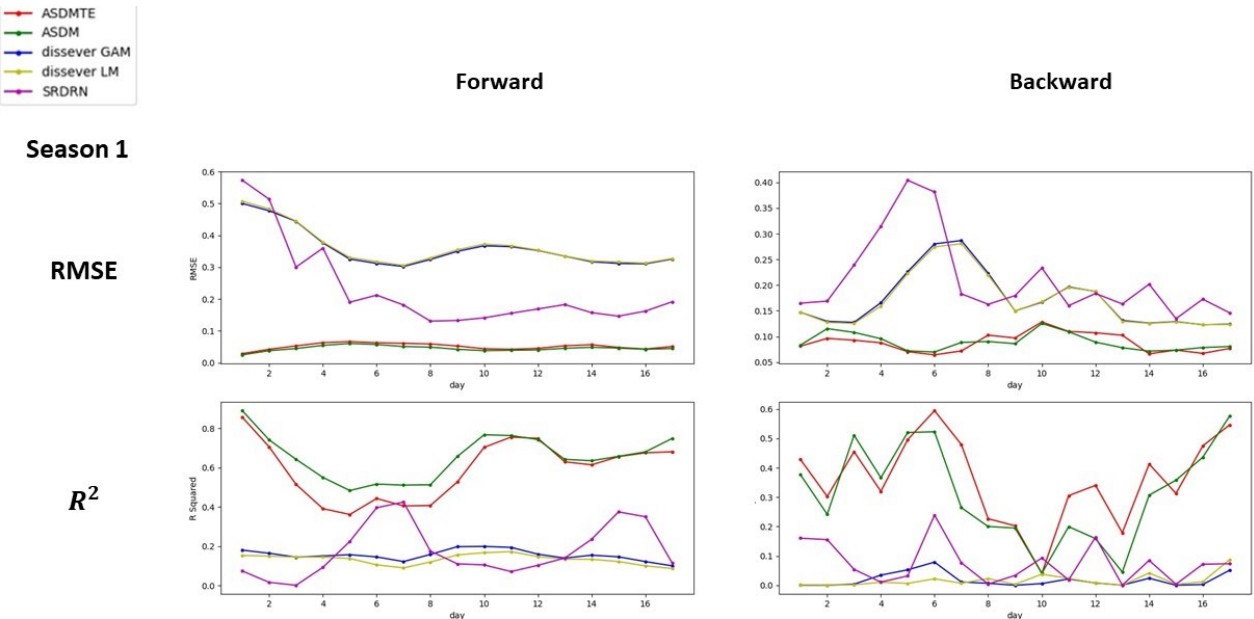

**Figure A1.** Downscaling performance of ASDM, ASDMTE, SRDRN, dissever GAM and dissever LM in Season 1. Please refer to Figure 4 and Section 2.2.3 for the definition of Season 1.

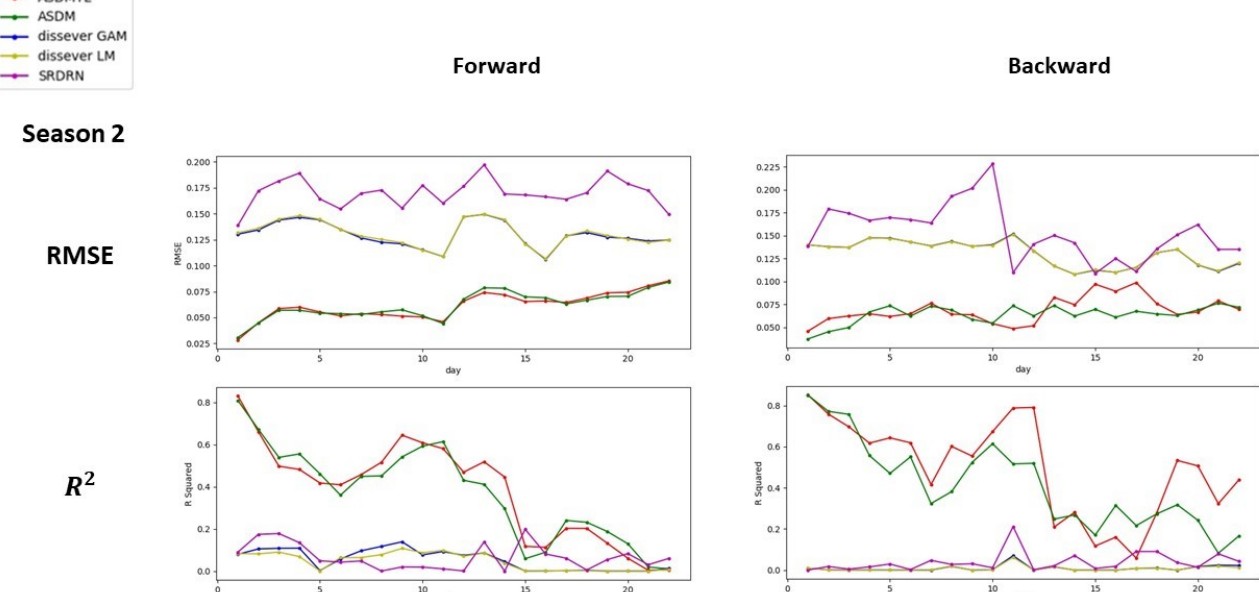

**Figure A2.** Downscaling performance of ASDM, ASDMTE, SRDRN, dissever GAM and dissever LM in Season 2. Please refer to Figure 4 and Section 2.2.3 for the definition of Season 2.

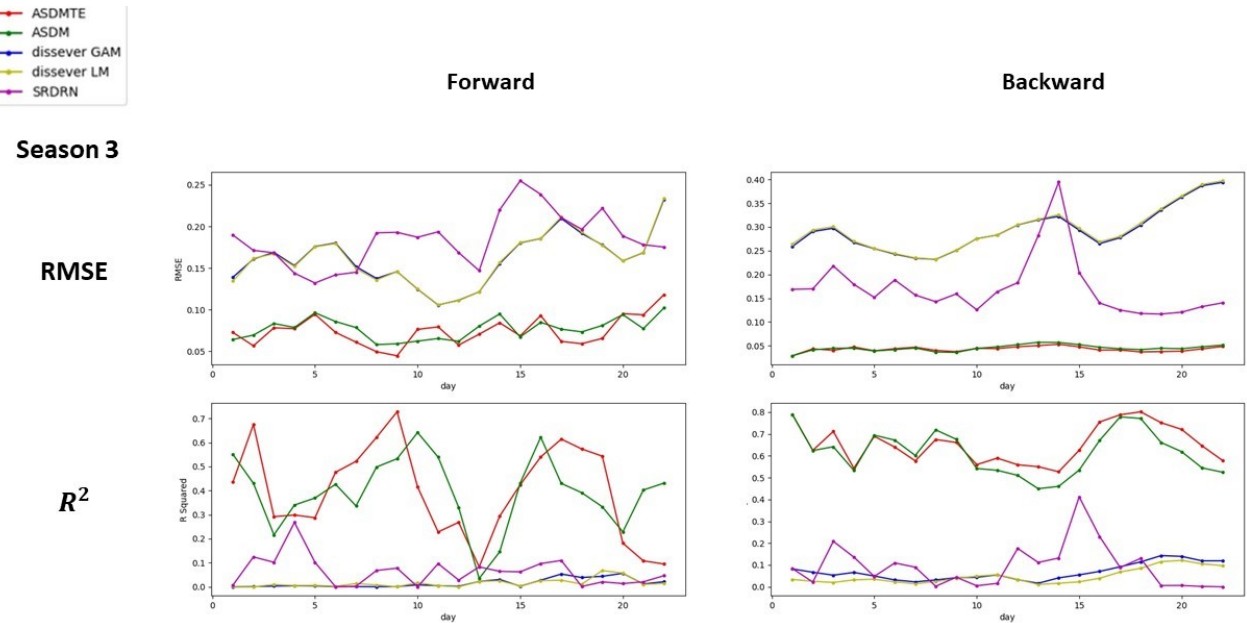

**Figure A3.** Downscaling performance of ASDM, ASDMTE, SRDRN, dissever GAM and dissever LM in Season 3. Please refer to Figure 4 and Section 2.2.3 for the definition of Season 3.

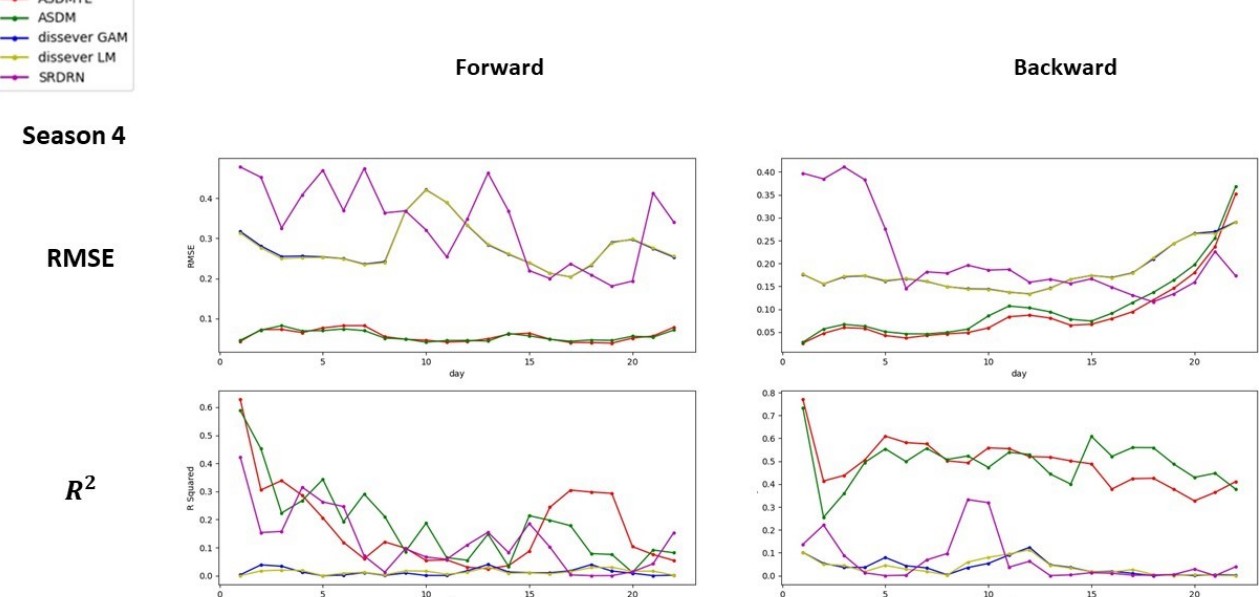

**Figure A4.** Downscaling performance of ASDM, ASDMTE, SRDRN, dissever GAM and dissever LM in Season 4. Please refer to Figure 4 and Section 2.2.3 for the definition of Season 4.

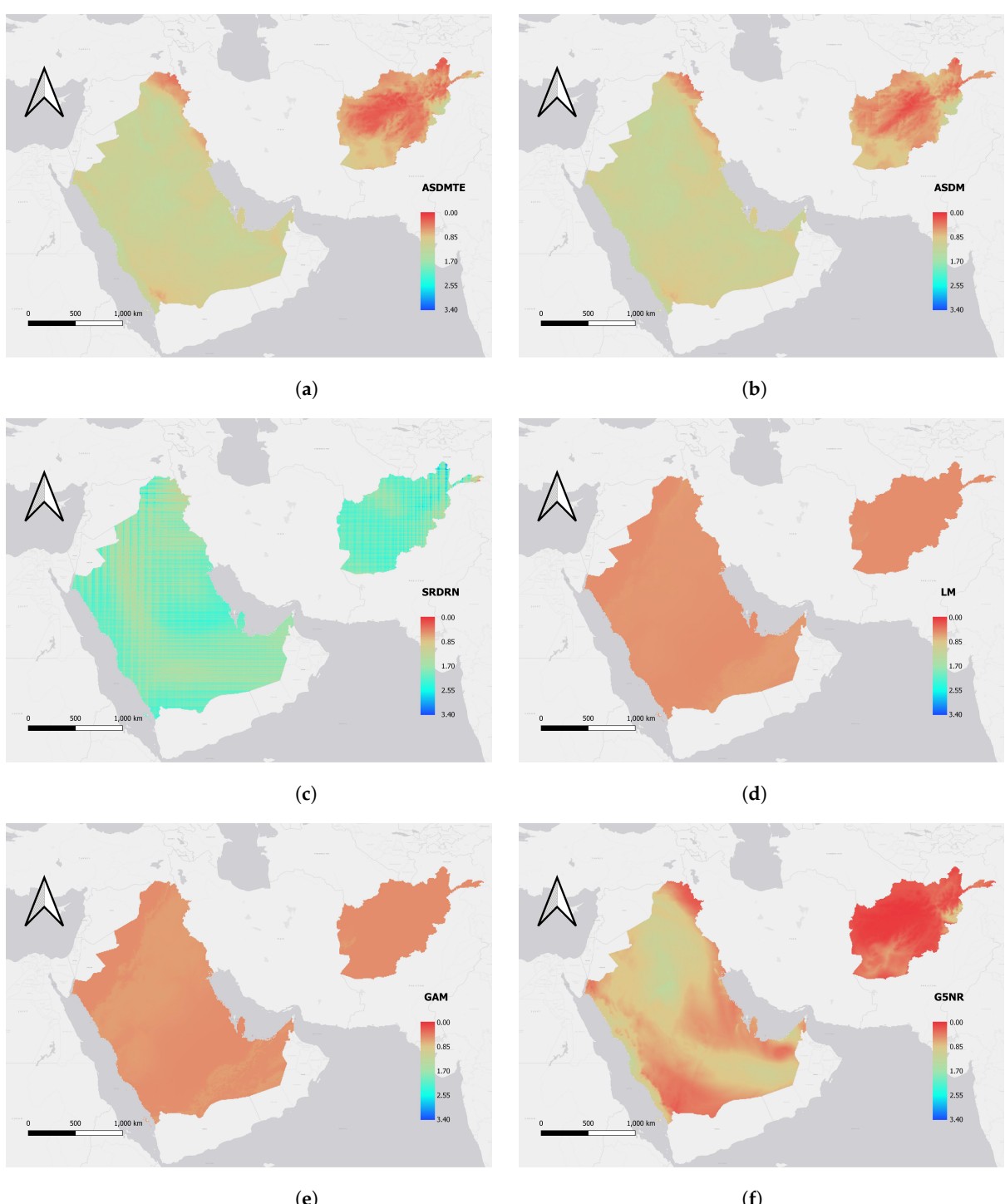

**Figure A5.** Downscaled (by method) and G5NR data over the study region on 23 October 2006: (**a**) ASDMTE, (**b**) ASDM, (**c**) SRDRN, (**d**) dissever GAM, (**e**) dissever LM, (**f**) G5NR.

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
