# Peer review of "Generating Fine-Scale Aerosol Data through Downscaling with an Artificial Neural Network Enhanced with Transfer Learning"

_atmosphere, doi:10.3390/atmos13020255_

Round 1
Reviewer 1 Report
This paper mainly studies from the perspective of deep learning method, and applies the method to the downscaling study of AOD from coarse-scale to fine-scale, which has potential application value for certain spatial and time series data.
The specific comments are as follows:
(1) In the introduction part, the progress of making high-resolution aerosol products based on multi-source satellite remote sensing data needs to be supplemented.
(2)The motivation of research area selection is not well analyzed in the following paragraphs.
(3) There are many satellite and model data related to aerosols. Why are these AOD data selected for downscaling study in this paper?
(4) The black shading and colorbar selection of Fig.5 and Fig.6 in this paper did not significantly highlight the difference.
(5) What does (0,6) indicate in line 118 and 131?
Author Response
Re: Resubmission of manuscript Generating Fine-Scale Aerosol Data Through Downscaling with an Artificial Neural Network Enhanced with Transfer Learning, atmosphere-1558287
January 28, 2022
Dear Reviewer:
Thank you for the opportunity to revise our manuscript, Generating Fine-Scale Aerosol Data Through Downscaling with an Artificial Neural Network Enhanced with Transfer Learning. We appreciate the careful review and constructive suggestions. We feel that the manuscript is substantially improved after making your suggested edits.
Following this letter are the editor and reviewer comments with our responses in italics, including how and where the text was modified. Changes made in the manuscript are highlighted in the attached PDF file.
We greatly appreciate your consideration.
Sincerely,
Meredith Franklin Menglin Wang
Associate Professor, Statistics Ph.D. Student, Biostatistics
University of Toronto University of Southern California
meredith.franklin@utoronto.ca menglinw@usc.edu
+1 (617) 877-1289 +1(213) 234-8950
Comments and Responses:
- In the introduction part, the progress of making high-resolution aerosol products based on multi-source satellite remote sensing data needs to be supplemented.
Thank you. We agree with the reviewer and have added text that provides more detail on MERRA-2 and G5NR AOD (line 44-58). We also added additional detail about MERRA-2 and G5NR in the methods section (line 134-136)
- The motivation of research area selection is not well analyzed in the following paragraphs.
Thank you. We agree with the reviewer and have added motivation for our research area selection in the Introduction section (line 35-43). We moved this text up so that it is earlier in the introduction section to motivate the reader earlier.
- There are many satellite and model data related to aerosols. Why are these AOD data selected for downscaling study in this paper?
Thank you. We agree with the reviewer and have added the motivation for why we used MERRA-2 and G5NR AOD data for our study in the Introduction section (line 44-58). Specifically, these data provide complete global surfaces of AOD and other aerosol products that are useful for downstream applications including epidemiological studies of the association between air pollution and health.
- The black shading and color bar selection of Fig.5 and Fig.6 in this paper did not significantly highlight the difference.
Thank you for these observations and we agree with the reviewer. We have updated Fig. 5 and Fig. 6 in the Result section, as well as Fig. 5 in Appendix, using light base-map and wide-scale color bar to better show the difference.
- What does (0,6) indicate in line 118 and 131?
Thank you. (0, 6) was the numerical range of AOD data, but since it is not nontrivial for the content, we have deleted them from line 118 and 131.

Reviewer 2 Report
The authors report on artificial neural network sequential downscaling method (ASDM) with transfer learning enhancement (ASDMTE) to downscale Aerosol Optical Depth (AOD) from coarse grain (CS – 50 km) satellite data to finer-scale results (FS- 7 km). The ASDM and ASDMTE downscaling schemes were applied to G5NR and MERRA-2 data (NASA GEOS-5 satellite data) for 6 middle eastern countries and compared with the results from other methods. Within scale temporal associations was modeled using Long Short Term Memory (LSTM). A Mean Square Error method was used as a loss function. The transferred model was trained on the CS data from MERRA-2 and random selection of 10% of the days was used for validation for early stopping and 32 models were fitted on all combinations of region, season and direction. The authors nicely illustrate that the ASDM and ASDMTE outperformed the other methods across all seasons and directions.
The authors discuss their approach in comparison with other statistical approaches such as linear regression and deep learning which ignore temporal associations in the data and illustrate the inherent with-scale temporal associations that exist in the FS and CS data. They also point out inherent weaknesses of down-scaling methods such as stationarity, test robustness and the requirement of FS data.
Overall, the paper is very well-written, well-organized and the results are presented in concise form. All tables and figures are of high quality. Although the subjective area is not in my area of expertise, I found the authors presented convincing and factual arguments that demonstrate the superior nature of their approach for the accurate downscaling from coarse to fine scale data. I feel this approach will be of wide interest the Atmosphere community and recommend publication as is.
Author Response
Re: Resubmission of manuscript Generating Fine-Scale Aerosol Data Through Downscaling with an Artificial Neural Network Enhanced with Transfer Learning, atmosphere-1558287
January 28, 2022
Dear Reviewer:
Thank you for the opportunity to revise our manuscript, Generating Fine-Scale Aerosol Data Through Downscaling with an Artificial Neural Network Enhanced with Transfer Learning. We appreciate the careful review and constructive suggestions. We feel that the manuscript is substantially improved after making your suggested edits.
Following this letter are the editor and reviewer comments with our responses in italics, including how and where the text was modified. Changes made in the manuscript are highlighted in the attached PDF file.
We greatly appreciate your consideration.
Sincerely,
Meredith Franklin Menglin Wang
Associate Professor, Statistics Ph.D. Student, Biostatistics
University of Toronto University of Southern California
meredith.franklin@utoronto.ca menglinw@usc.edu
+1 (617) 877-1289 +1(213) 234-8950
Comments and Responses:
The authors report on artificial neural network sequential downscaling method (ASDM) with transfer learning enhancement (ASDMTE) to downscale Aerosol Optical Depth (AOD) from coarse grain (CS – 50 km) satellite data to finer-scale results (FS- 7 km). The ASDM and ASDMTE downscaling schemes were applied to G5NR and MERRA-2 data (NASA GEOS-5 satellite data) for 6 middle eastern countries and compared with the results from other methods. Within scale temporal associations was modeled using Long Short Term Memory (LSTM). A Mean Square Error method was used as a loss function. The transferred model was trained on the CS data from MERRA-2 and random selection of 10% of the days was used for validation for early stopping and 32 models were fitted on all combinations of region, season and direction. The authors nicely illustrate that the ASDM and ASDMTE outperformed the other methods across all seasons and directions.
The authors discuss their approach in comparison with other statistical approaches such as linear regression and deep learning which ignore temporal associations in the data and illustrate the inherent with-scale temporal associations that exist in the FS and CS data. They also point out inherent weaknesses of down-scaling methods such as stationarity, test robustness and the requirement of FS data.
Overall, the paper is very well-written, well-organized and the results are presented in concise form. All tables and figures are of high quality. Although the subjective area is not in my area of expertise, I found the authors presented convincing and factual arguments that demonstrate the superior nature of their approach for the accurate downscaling from coarse to fine scale data. I feel this approach will be of wide interest the Atmosphere community and recommend publication as is.
We greatly appreciate your comments and we have made the following changes to improve our manuscript:
- We added text that provides more detail on MERRA-2 and G5NR AOD (line 44-58). We also added additional detail about MERRA-2 and G5NR in the methods section (line 134-136)
- We have added motivation for our research area selection in the Introduction section (line 35-43). We moved this text up so that it is earlier in the introduction section to motivate the reader earlier.
- We have added the motivation for why we used MERRA-2 and G5NR AOD data for our study in the Introduction section (line 44-58). Specifically, these data provide complete global surfaces of AOD and other aerosol products that are useful for downstream applications including epidemiological studies of the association between air pollution and health.
- We have updated Fig. 5 and Fig. 6 in the Result section, as well as Fig. 5 in Appendix, using light base-map and wide-scale color bar to better show the difference.
- We have deleted (0, 6) from line 118 and 131. (0, 6) was the numerical range of AOD data, but since it is not nontrivial for the content, we deleted them to keep the manuscript concise.
